# CTQScorer: Combining Multiple Features for In-context Example Selection for Machine Translation

**Aswanth Kumar**[1] and **Ratish Puduppully**[2] and **Raj Dabre**[3] and **Anoop Kunchukuttan**[4]

IIT Madras, India[1,4]    Microsoft, India[4]
Institute for Infocomm Research (I[2]R), A*STAR, Singapore[2]    AI4Bharat[4]
National Institute of Information and Communications Technology, Kyoto, Japan[3]
[1]kumaraswanth@gmail.com   [2]puduppully@i2r.a-star.edu.sg
[3]raj.dabre@nict.go.jp   [4]ankunchu@microsoft.com

## Abstract

Large language models have demonstrated the capability to perform on machine translation when the input is prompted with a few examples (*in-context learning*). Translation quality depends on various features of the selected examples, such as their quality and relevance, but previous work has predominantly focused on individual features in isolation. In this paper, we propose a general framework for combining different features influencing example selection. We learn a regression model, *CTQ Scorer* (Contextual Translation Quality), that selects examples based on multiple features in order to maximize the translation quality. On multiple language pairs and language models, we show that CTQ Scorer helps significantly outperform random selection as well as strong single-factor baselines reported in the literature. We also see an improvement of over 2.5 COMET points on average with respect to a strong BM25 retrieval-based baseline.

## 1 Introduction

Large language models (LLMs) trained on massive amounts of textual data have demonstrated impressive performance on a wide range of NLP tasks despite not being explicitly trained on any of them (Liu et al., 2023; Chung et al., 2022; Goyal et al., 2022b; Wei et al., 2022; Chowdhery et al., 2022). These capabilities of the model are elicited using *in-context learning*, where the model is prompted with task instructions and demonstration examples followed by the input. The task's output for the given input is simply the next sequence of tokens sampled from the language model.

Recently, in-context learning has also been explored for machine translation (Brown et al., 2020; Chowdhery et al., 2022; Lin et al., 2022; Scao et al., 2022). Many of these models have shown encouraging results for translation, particularly for high-resource languages. This achievement is impressive given that the models have not been inten-

tionally supplied with parallel data, and their training data predominantly consists of English content. However, performance on low-resource languages and translation out of English is an unresolved major challenge, where issues like hallucination and adequacy gaps have been observed. On the other hand, LLM translations are more fluent and paraphrastic, and they handle long-distance reordering better - especially when translating into English (Hendy et al., 2023).

An important aspect of in-context learning is the creation of the prompt. The prompt typically consists of two parts: the *prompt template* that helps the model understand the task and the *in-context examples* that aid in the better translation of the *input source* sentence. The in-context examples can be selected for each input source from an *example database/datastore* that contains parallel sentence pairs. The number, order, and choice of examples can affect the translation quality (Zhang et al., 2023). Specifically, various features of examples like the translation quality of the sentence pair, the length of sentences, its semantic similarity to the input, *etc.* can contribute to the translation quality.

Previous approaches consider only individual features when selecting examples. However, such an approach could be sub-optimal as it ignores the relevance of other features to the translation quality. For instance, assume that we choose examples based on the quality of the translation pairs. The selected sentences may be short, potentially offering limited information for the translation of the input. Moreover, the selection depends on the chosen translation quality metric. Given that translation quality metrics are imperfect, a different metric might have led to a different selection of examples. A better approach would be to select examples based on a diverse set of features and different views of measuring the same feature (*e.g.* translation quality could be measured by various metrics such as LaBSE (Feng et al., 2020) or COMET-QE

(Rei et al., 2020b)) to maximize translation quality.

In this work, we explore example selection based on multiple features. Our contributions are:

**1.** We propose selecting examples based on a scoring function that integrates evidence from different features. We propose *CTQ Scorer*, a regression-based scoring function that estimates the translation quality given the example and the test input.

**2.** Given the absence of manually annotated quality scores for training the proposed regression model, we propose a novel method for creating training data. We estimate *contextual translation quality* on a held-out set by *1-shot* prompting on the LLM using the input source from the held-out set with an example from the database as context. These (input source, example, quality score) tuples serve as training data for the regression model.

**3.** We show that combining evidence from multiple features selects examples that improve translation quality compared to the use of individual features. CTQ Scorer helps improve translation quality on multiple language pairs and language models.

**4.** In addition to measures used in the past for example selection, we explore new features. Based on our study of various features used, we find that: (a) COMET-QE features (learned metrics) are better at example selection than cosine similarity of LaBSE embeddings (task-agnostic semantic matching), (b) Similarity of the source input to the example target is more important than its similarity to example source, (c) combining the two observations into a novel feature *i.e.* using COMET-QE based similarity between input source and target is the best feature and is a strong baseline across languages and directions.

Our code and outputs are accessible via this repository: https://github.com/AI4Bharat/CTQScorer

## 2 Related Work

The selection of an appropriate prompt to enhance machine translation (MT) performance of large language models has been the focus of recent research (Zhang et al., 2023; Li et al., 2022; Agrawal et al., 2022; Liu et al., 2021; Jiang et al., 2020; Zemlyanskiy et al., 2022; Rubin et al., 2021).

The relevance of the examples to the input sentences is an important factor that affects translation quality. Several methods for measuring relevance have been employed: (a) n-gram word overlap between input sentence and examples (Vilar et al., 2022; Agrawal et al., 2022), and (b) embedding similarity (Zhang et al., 2023; Vilar et al., 2022; Hendy et al., 2023) using LaBSE (Feng et al., 2022) or RoBERTa (Liu et al., 2019) embeddings. The quality of the examples is also an important factor. To ensure quality, examples are either selected from a known high-quality pool (Vilar et al., 2022) or based on LaBSE or COMET-QE (Rei et al., 2020b) scores between the pairs (Zhang et al., 2023; Hendy et al., 2023). Sia and Duh (2023) show that coherency of prompt examples with respect to the test sentence is an important factor for translation performance.

Agrawal et al. (2022) further explore task-level examples, *i.e.* fixed high-quality examples that result in the best translation quality on a held-out set. Zhang et al. (2023) study various factors affecting example selection, demonstrating that these features exhibit weak correlation to translation quality, and no single feature can consistently enhance translation quality.

However, all these works select examples based on a single feature. In contrast, we propose to combine different features contributing to translation quality for a more informed example selection process. Additionally, we investigate some novel features that can influence example selection.

## 3 Example Selection using Multiple Features

Given an input sentence $x$ in source language $s$, we want to select a set of $k$ examples ($E$) from an example database ($D$) to aid in generating the best translation of $x$ into the target language $t$. In the case of MT, the example database corresponds to a parallel corpus comprising translation pairs $(x_p, y_p)$ from which the prompt examples are drawn. Our overall approach, as shown in Figure 1 is described in this section.

**1. Candidate Shortlisting** Initially, we identify $n$ examples $E = \{(x_p, y_p) \mid p = 1, 2, .., n\}$ from $D$ that are similar to the input sentence $x$. These $n$ candidates are subsequently re-ranked based on multiple features for the final selection of $k$ in-context examples. Following Agrawal et al. (2022), we employ BM25[1], an unsupervised efficient retriever, to locate these similar examples. This method ensures that the selected examples exhibit high n-gram word overlap with the input source.

---

[1] We used this implementation of BM25 retrieval: https://github.com/AmenRa/retriv

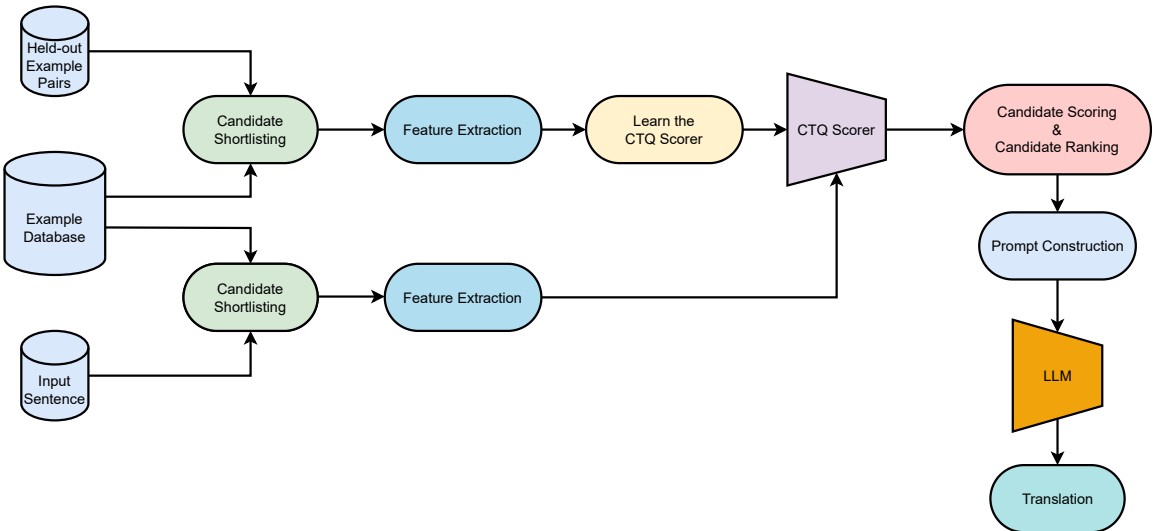

Figure 1: Overview of our LLM-based example selection system. The system selects in-context examples by incorporating multiple features to estimate the Contextual Translation Quality (CTQ) score. The upper part of the figure illustrates the training process of the CTQ Scorer. Candidate shortlisting is conducted using held-out example pairs, and feature extraction is performed on these shortlisted candidates. The extracted features are used for the training of the CTQ Scorer. During the inference stage, as depicted in the lower part of the figure, candidate shortlisting is performed for an input sentence from the examples in the example database, and relevant features are extracted. The CTQ Scorer then performs candidate scoring and ranking based on these extracted features. Subsequently, the best examples are chosen based on the CTQ score, and a prompt is constructed. Finally, the LLM produces the machine translation output using this constructed prompt.

An alternative could be to identify $n$ examples $(x_p)$ nearest to $x$ in the embedding space for a better semantic match. However, we opted for the BM25 retriever for efficiency reasons.

**2. Feature Extraction** For each candidate $(x_p, y_p)$ and input $x$, we extract a range of features featset$(x_p, y_p, x)$ that could impact translation quality for $x$. The specific features utilized are described in Section 3.2.

**3. Candidate Scoring** The extracted features are used by an example scoring function, ctx_xlate_score$(x_p, y_p, x)$, *Contextual Translation Quality* Scorer (CTQ Scorer) which assigns a score to each candidate example. The CTQ scorer predicts the translation quality of the input $x$ into the target language, given the *single* candidate example $(x_p, y_p)$ as context during prompting. The scoring function and its learning process are further elaborated in Section 3.1.

**4. Candidate Ranking** The candidates are ranked according to their CTQ score, with the top $k$ candidate examples being selected for $k$-shot prompting.

**5. Translation** The prompts are constructed using the specified instruction template, the $k$ selected

examples and input $x$. The LLM is then prompted to generate a completion for the prompt, with the completion $(y')$ serving as the generated translation for the input $x$.

### 3.1 Contextual Translation Quality Scorer

Our goal is to select examples that can maximize the translation quality of an input sentence given the examples as context—a measure we term as the *Contextual Translation Quality* (CTQ). The CTQ score is computed as a function of the features extracted from the example and the input source $(x_p, y_p, x)$. Notably, it does not depend on the translation output. This approach is necessary in order to assign a score for selection without requiring translations of the input conditioned on the examples from the LLM. Essentially, the CTQ score is an estimate of the translation quality that the in-context example can provide.

We model the CTQ scorer as a regression function that outputs a scalar CTQ score given features extracted from the $(x_p, y_p, x)$ tuple. In the typical training of translation quality estimation models, human judgment scores are available. However, in this case, we lack human judgments for CTQ. Consequently, we propose an approach for generating training data for the regression model.

**Algorithm 1** Algorithm for creation of data to train the CTQ Scorer regression model

---
1: **Inputs** Held-out example pairs *(x, y)*, example database *D*
2: **Outputs** Training data for CTQ Scorer regression model
3: **procedure** CREATETRAININGDATA
4:     **for** a given $(x, y)$ from held-out example pairs **do**
5:         Perform *Candidate Shortlisting* and retrieve $K$ candidate examples from *D*
6:         Each of the tuple $(x_p, y_p)$ in *K* candidate examples is a prompt candidate
7:         **for** a given $(x_p, y_p)$ **do**
8:             Generate the 1-shot translation $y'$ of $x$ using $(x_p, y_p)$ as prompt example
9:             Generate the Translation score using any sentence-level MT metric, xlate_score$(x, y', y)$
10:             ctq = xlate_score$(x, y', y)$
11:             featset$(x_p, y_p, x)$ = *Feature Extraction* using the triple $(x_p, y_p, x)$
12:             *Training Instance* = featset$(x_p, y_p, x)$, ctq
13:     **return** *All Training Instances*

---

Assume we have a small held-out parallel corpus of $N$ $(x \in X, y \in Y)$ sentence pairs. For a given source $x$ from a held-out sentence pair, we retrieve $K$ candidate examples using a BM25 retriever, as discussed earlier. With each candidate $(x_p, y_p)$ as in-context example, we sample a translation $y'$ as the LLM output for $x$. We then compute the translation quality (CTQ) of $(x, y')$ using the reference $y$ with any sentence-level MT evaluation metric (xlate_score$(x, y', y)$). Specifically, in this work, we use COMET (*wmt20-comet-qe-da*) to compute CTQ. As a result, we obtain $(x_p, y_p, x, \text{CTQ})$ as training instances for the regression model. We can train the CTQ scorer regression model on this synthetic training data. Note that the CTQ model is trained to score each example independently. Algorithm 1 outlines the process of training data creation.

## 3.2 Features used by the CTQ Scorer

In order to estimate the CTQ score, we use several features relevant to example selection, extending the list mentioned by Zhang et al. (2023). We consider the following features:

### 3.2.1 Example Similarity Features

These features measure the semantic similarity between the example and the input. We consider similarity between example source and input as well as example target and input. We incorporate multiple metrics of similarity, encompassing cosine similarity of embeddings, lexical metrics, and QE metrics. The application of a QE metric for determining semantic match is a unique approach in this context.

**LaBSE-InSrc:** The cosine similarity between the input and source of the example sentence, computed using LaBSE embeddings (Feng et al., 2022).

**LaBSE-InTgt:** The cosine similarity between the input and target of the example sentence, computed using LaBSE embeddings.

**chrF-InSrc:** The chrF score (Popović, 2015) is a MT evaluation metric that uses the F-score statistic for character n-gram matches. We computed the chrF score between the input and source side of the example sentence.

**Cmt-InSrc:** The COMET-QE score between the input and source of the example sentence.

**Cmt-InTgt:** The COMET-QE score between the input and target of the example sentence.

### 3.2.2 Example Quality Features

**LaBSE-SrcTgt:** This is the cosine similarity score between the source and target of the example sentence, computed using LaBSE embeddings. This is indicative of the translation quality of the example.

**Cmt-SrcTgt:** We also evaluated the COMET-QE score of source and target of the example. This score measures the translation quality of the in-context example.

### 3.2.3 Other Features

**NumTokIn:** The number of tokens in the input.

**NumTokSrc:** The number of tokens in the source side of the example.

**NumTokTgt:** The number of tokens in the target side of the example.

**PPL-SrcTgt** and **PPL-SrcTgtIn:** We explore two features related to the perplexity of the example: (a) the perplexity of the concatenated source and target of the example, and (b) perplexity of the source, target, and input concatenated. The perplexities are computed on the same LLM that is used for translation. These features are inspired by Gonen et al. (2022) who show that language models are likely to perform well on prompts they are familiar with. Lower perplexity indicates higher familiarity of the model to the prompt. Our application of

these features differs from Gonen et al. (2022) in the following aspects: (a) they use this feature for prompt template selection while we use it for example selection, (b) they address classification tasks while we focus on a generation task *viz.* translation.

Of the features described above, the novel features we proposed in this work are: chrF-InSrc, Cmt-InSrc, Cmt-InTgt, PPL-SrcTgt and, PPL-SrcTgtIn. While we could use multiple instances of the same metric (*e.g.* LaBSE, LASER for cosine similarity; BLEU, chrF for lexical metrics; BLEURT, BERTScore for learned metrics), we chose to limit our feature set initially to demonstrate the utility of multi-feature example selection. Incorporation of multiple views of the same metric, novel features, etc. can be easily included in our framework, and we will explore this in future experiments. We use the *wmt20-comet-qe-da* model for COMET-QE computation.

## 3.3 CTQ Scorer Model

Our model CTQ Scorer is a language-specific neural regression model comprising an input layer, hidden layers, and an output layer. The number of neurons in the input layer corresponds to the features present in $\text{featset}(x_p, y_p, x)$, while the output layer contains a single neuron, as we predict the CTQ Score. The hidden layers apply non-linear transformations to the input data, enabling the model to learn intricate relationships between the extracted features and the CTQ Score. The CTQ Scorer's parameters are optimized through learning from the training data.

## 4 Experimental Setup

We conducted transalation experiments between English and Bengali, Gujarati, Hindi, French, German, and Russian. We also studied example selection using different selection algorithms, along with the Contextual Translation Quality (CTQ) Scorer approach discussed in Section 3.1.

| Language | ISO code | Dataset | #Pairs (M) |
|----------|----------|---------|-----------|
| Bengali | bn | Samanantar | 8.6 |
| Gujarati | gu | Samanantar | 3.1 |
| Hindi | hi | Samanantar | 10.1 |
| French | fr | Europarl | 1.9 |
| German | de | Europarl | 1.8 |
| Russian | ru | ParaCrawl | 5.4 |

Table 1: Example datasets and the number of sentence pairs per language (in millions).

## 4.1 Datasets

**Example Database:** The example database consists of parallel sentences from Samanantar (Ramesh et al., 2022), Europarl (Koehn, 2005), and Paracrawl (Bañón et al., 2020). Detailed statistics on the size of the example database for each language pair can be found in Table 1.

**Generating training data for the CTQ scorer:** Using Algorithm 1 we use the *dev* set of FLORES-101 (Goyal et al., 2022a), containing $N = 997$ sentence pairs, as the held out data along with the example database to create training data for the CTQ Scorer. We use $K = 100$ retrieved examples per input sentence in the *dev* set. Each retrieved example is used for 1-shot prompting and this leads to 99,700 training instances which are divided into an 8:1:1 ratio for training, validation and testing. Note that, we use 1-shot prompting for training data generation because we want to score each example independently for reranking.

**Evaluation data:** We report scores on the *devtest* set of FLORES-101 (Goyal et al., 2022a).

## 4.2 Evaluation Metrics

The primary evaluation metric utilized in our experiments is COMET (Rei et al., 2020a), calculated using the *wmt20-comet-da* model, since the recent WMT Metrics Shared Task (Freitag et al., 2022) finds that neural fine-tuned metrics are better and robust to domain-shift. For completeness, we also report BLEU (Papineni et al., 2002) and other metric scores in Appendix B.

## 4.3 Prompting Setup

**LLMs used:** The experiments were conducted utilizing the BLOOM 7.1B model (Scao et al., 2022) and the XGLM 7.5B model (Lin et al., 2022). Both the models are multilingual generative language models which are trained on a corpus covering a diverse set of languages and has few-shot learning capabilities on a wide range of tasks. BLOOM supports 11 Indic languages and XGLM supports German, French and Russian languages.

| Lang | Model | #Param | Layers | Model Dim |
|------|-------|--------|--------|-----------|
| bn, gu, hi | BLOOM | 7.1B | 30 | 4096 |
| de, fr, ru | XGLM | 7.5B | 32 | 4096 |

Table 2: Languages along with the corresponding models utilized for evaluating MT performance.

**Prompt Template:** In order to ensure comparable

results across all experiments, a fixed prompt template was used for *k-shot* prompting. The template takes the following form:

```
[source] sentence: [X_1]
[target] sentence: [Y_1]
###
...
[source] sentence: [X_k]
[target] sentence: [Y_k]
###
[source] sentence: [X]
[target] sentence:
```

Within this template, [source] and [target] are placeholders that are replaced with the names of the source and target languages in English, such as Hindi and English. The ### symbol is used as an example delimiter and is used as a marker for post-processing.

**Pre-Processing and Post-Processing:** We pre-process the in-context examples to eliminate duplicates and those that cause the context to exceed 1000 tokens. *We observed that the elimination of duplicates is important to ensure that the BM25 retriever retrieves high-quality, diverse examples from the example store.* Since the LLM is unable to know when to stop generating, we eliminate all text after the delimiter ('###') encountered in the generated output.

### 4.4 CTQ Scorer Configuration

We discovered the optimal CTQ Scorer model for each language pair/direction combination by hyperparameter search over the number of hidden layers, number of neurons in the hidden layer, learning rate, optimizer algorithm, activation function, batch size, and weight decay, ensuring a minimized validation set error. For optimization, we utilized algorithms such as stochastic gradient descent (SGD) (Robbins and Monro, 1951), Adam (Kingma and Ba, 2014), and RMSProp (Tieleman et al., 2012). We employed Mean Squared Error (MSE) as the loss function. Detailed information along with the optimal configuration is provided in the Appendix in Table 8 and Table 10.

### 4.5 Example Selection Methods Compared

We compared the following methods for selection of $k$ in-context examples with $k = 4$.

**Random Selection:** Examples are selected randomly for each test input from the example database. We report the average results of three runs with random selection for evaluating MT quality. The performance was assessed by averaging the scores obtained from three different seeds.

**BM25:** We compare with the approach described by Agrawal et al. (2022), where $k$ examples are retrieved such that the source sentences in the example database are most similar to the test source. The match is performed using BM25 retriever, which focuses on n-gram word overlap.

The other baselines follow a re-ranking approach. Initially, they use the BM25 retriever to extract the top-100 matching examples. These examples are then re-ranked according to varying criteria. The top-$k$ reranked examples are used to prompt the LLM.

**R-BM25:** This baseline replicates the reranking algorithm implemented in Agrawal et al. (2022) which aims to achieve greater overlap between the input source and examples by ensuring greater n-gram diversity in the top-$k$ examples.

**Individual Features:** We experiment with systems where examples are selected by reranking just one feature. We consider all the features described in Section 3.2.

**CTQ Scorer:** Our proposed method to select examples based on multiple features.

## 5 Results and Analysis

### 5.1 Main Results

The main results for translation into English and out of English are presented in Table 3 and Table 4[2] respectively.

**Example selection using the CTQ Scorer outperformed other methods.** We see that CTQ Scorer is the best method compared to all baselines and individual features in both directions. We observe a significant +2.5 and +4.5 COMET points gain on average in XE (translation into English) and EX (translation from English) directions respectively over the random baseline. CTQ also significantly improves over the BM25 baseline since it looks at many other factors in addition to just n-gram overlap. This trend holds for R-BM25 as well which promotes more word n-gram diversity in the selected examples. While we have not compared with finding the best matching examples based on embedding search over the entire example database

---

[2]en-gu results are not reported since COMET-20 seems badly calibrated for this pair. Results on COMET-22 in the Appendix show that en-gu trends are in line with observations for other languages.

| Selection Method | bn | gu | hi | de | fr | ru | Average |
|---|---|---|---|---|---|---|---|
| Random Selection | 40.07 | 38.27 | 44.52 | 63.05 | *70.89 | *49.40 | 51.03 |
| BM25 | 38.93 | 38.42 | 45.18 | 62.14 | *70.82 | 45.76 | 50.21 |
| R-BM25 | 39.97 | 38.16 | 45.20 | 62.94 | *70.31 | *49.28 | 50.98 |
| CTQ (*ours*) | **42.99** | **41.77** | **50.03** | **64.77** | 71.28 | 50.85 | **53.62** |
| CTQ-QE (*ours*) | 38.56 | 40.45 | 45.40 | 64.13 | **71.33** | 50.72 | 51.76 |
| *Individual Features* | | | | | | | |
| NumTokSrc | 38.02 | 39.53 | 42.23 | 61.88 | 70.57 | 47.50 | 49.96 |
| NumTokTgt | 39.44 | 35.06 | 39.33 | 61.95 | 70.66 | 45.38 | 48.64 |
| CmtQE-InSrc | 39.17 | 38.02 | 44.77 | 63.57 | 69.94 | 50.25 | 50.95 |
| CmtQE-InTgt | 40.33 | 40.28 | 48.07 | 64.76 | 69.82 | **51.15** | 52.40 |
| CmtQE-SrcTgt | 39.79 | 38.79 | 48.84 | 62.51 | 69.97 | 46.67 | 51.10 |
| chrF-InSrc | 37.63 | 38.08 | 43.41 | 59.82 | 69.49 | 40.13 | 48.09 |
| LaBSE-InSrc | 40.06 | 37.71 | 46.71 | 63.50 | 70.43 | 48.04 | 51.08 |
| LaBSE-InTgt | 41.30 | 38.70 | 47.36 | 61.96 | 70.49 | 44.65 | 50.74 |
| LaBSE-SrcTgt | 41.12 | 40.02 | 48.12 | 62.07 | 70.07 | 39.59 | 50.17 |
| PPL-SrcTgt | 40.52 | 40.91 | 45.39 | 63.62 | 70.87 | 47.22 | 51.42 |
| PPL-SrcTgtIn | 39.96 | 41.27 | 46.11 | 63.16 | 71.20 | 47.05 | 51.46 |
| *Comparison with Score Averaging* | | | | | | | |
| ScAvg (3-feat) | 42.75 | 41.20 | 48.35 | 62.63 | 70.61 | 43.99 | 51.59 |
| CTQ (3-feat) | 42.07 | 40.65 | 49.63 | 63.77 | 70.85 | 49.31 | 52.71 |

Table 3: COMET scores for translation into English using different example selection methods. The highest scores are in **bold** text. We compared CTQ with Random, BM25 and R-BM25 for statistical significance. All comparisons with CTQ are statistically significant (p<0.05) (except results marked with *) as per paired bootstrap sampling (Koehn, 2004).

.

due to computational reasons, re-ranking the BM25 retrieved results with embedding based features (LaBSE-InSrc, LaBSE-InTgt) is a reasonable substitute for the same. We can see that CTQ outperforms these approximations to embedding search based methods.

**Comparison with Example Quality Features.** We see that CTQ outperforms reranking based solely on example quality features (X-SrcTgt) according to both LaBSE and COMET-QE. This underscores the value added by the information from other features in the process of example selection. LaBSE-SrcTgt is particularly a weak feature for translating out of English. We hypothesize that since LaBSE looks at only semantic match based on embedding, and ignores other aspects of translation it is not able to select good-quality examples. COMET-QE based selection does not suffer from this limitation.

**Comparison with Example Similarity Features.** We see that CTQ outperforms reranking based on just example similarity features (X-InSrc and X-InTgt) based on LaBSE, COMET-QE and chrF. The similarity-based features perform better than the example quality features, but combining all features is still beneficial. We see that chrF brings only marginal benefits or causes regressions over the baseline BM25 method. Since chrF is a lexical metric, it probably does not add much information

to what BM25 provides. We also observe that similarity of the input source to the example target is more important than its similarity to the example source. This corroborates the findings of Zhang et al. (2023) on a wider set of languages.

**Observations on newly proposed features.**
• We observe that COMET-QE is a better example quality metric than LaBSE for example selection. Previous work has not compared it as a translation quality metric for example selection.[3]
• COMET-QE metrics are better than LaBSE for example similarity too, which has not been explored previously. In particular, we find that *CmtQE-InTgt* (matching input source with example target using COMET-QE) is the best-performing feature, and using this feature alone gives a very strong example selection method.
• We find that the perplexity features are useful and perform significantly better than the BM25 baselines in most cases. They complement the features discussed previously.

**Off-target translation.** We also studied if the models generate off-target translations (*i.e.* translation in a language other than the target language). We see that CTQ has amongst the lowest off-target translation rates. Detailed results are shown in Appendix C.

---

[3]Hendy et al. (2023) mention using LaBSE after comparison with COMET, but no results are reported.

| Selection Method | bn | hi | de | fr | ru | Average |
|---|---|---|---|---|---|---|
| Random Selection | 21.19 | 30.77 | 34.07 | *40.69 | 33.55 | 32.05 |
| BM25 | *23.96 | 28.16 | 35.04 | *41.57 | 37.60 | 33.27 |
| R-BM25 | *24.52 | *30.79 | *36.80 | *41.70 | 39.59 | 34.68 |
| CTQ (*ours*) | 26.02 | 33.36 | **38.05** | 41.41 | 44.26 | 36.62 |
| CTQ-QE (*ours*) | 25.92 | 34.25 | 35.98 | 42.37 | 42.32 | 36.17 |
| *Individual Features* | | | | | | |
| NumTokSrc | 31.70 | 28.56 | 29.99 | 36.26 | 35.20 | 32.34 |
| NumTokTgt | **32.04** | 27.26 | 31.68 | 35.04 | 35.70 | 32.34 |
| CmtQE-InSrc | 25.51 | 30.81 | 37.44 | 43.72 | 39.92 | 35.48 |
| CmtQE-InTgt | 26.56 | **35.13** | 37.84 | **44.38** | **44.46** | **37.67** |
| CmtQE-SrcTgt | 26.65 | 30.75 | 36.09 | 40.27 | 40.57 | 34.87 |
| chrF-InSrc | 24.11 | 28.96 | 34.53 | 38.77 | 36.73 | 32.62 |
| LaBSE-InSrc | 24.32 | 29.89 | 33.25 | 40.91 | 39.39 | 33.55 |
| LaBSE-InTgt | 28.45 | 33.72 | 35.52 | 38.85 | 40.93 | 35.49 |
| LaBSE-SrcTgt | 23.01 | 25.50 | 32.73 | 35.37 | 31.76 | 29.67 |
| PPL-SrcTgt | 30.87 | 31.72 | 36.39 | 42.28 | 37.71 | 35.79 |
| PPL-SrcTgtIn | 28.69 | 32.60 | 31.42 | 36.28 | 37.60 | 33.32 |
| *Comparison with Score Averaging* | | | | | | |
| ScAvg (3-feat) | 26.62 | 35.03 | 34.36 | 41.08 | 39.42 | 35.30 |
| CTQ (3-feat) | 27.72 | 34.63 | 36.17 | 42.14 | 41.88 | 36.51 |

Table 4: COMET scores for translation out of English using different example selection methods. The highest scores are in **bold** text. Statistical significance protocol is the same as in Table 3.

| Pair | bn-en | gu-en | hi-en | en-bn | en-gu | en-hi |
|---|---|---|---|---|---|---|
| **Neural** | **42.99** | **41.77** | **50.03** | 26.02 | **4.15** | 33.36 |
| **Linear** | 40.88 | 40.70 | 49.90 | **28.27** | 2.15 | **34.30** |

Table 5: COMET scores for comparing Neural and Linear Regression. The highest scores are in **bold** text.

| Pair | bn | gu | hi | fr | de | ru | Avg. |
|---|---|---|---|---|---|---|---|
| **W2B** | [†]**42.99** | **41.77** | **50.03** | **71.28** | **64.77** | [†]**50.85** | **53.62** |
| **B2W** | 39.87 | 41.28 | 49.64 | 71.13 | 63.86 | 47.63 | 52.24 |

Table 6: COMET scores comparing ordering of examples based on their CTQ scores while translating into English. W2B: Worst to Best ordering, B2W: Best to Worst ordering. The highest scores are in **bold** text. Scores that are statistically significantly better ($p < 0.05$) (marked with [†]) are obtained as per paired bootstrap sampling (Koehn, 2004).

| Feature | Importance coefficient |
|---|---|
| LaBSE-InSrc | 0.315 |
| PPL-SrcTgt | 0.304 |
| LaBSE-SrcTgt | 0.265 |
| chrF-InSrc | 0.048 |
| CmtQE-InTgt | 0.029 |
| CmtQE-SrcTgt | -0.002 |
| CmtQE-InSrc | -0.047 |
| NumTokIn | -0.054 |
| LaBSE-InTgt | -0.223 |
| NumTokTgt | -0.252 |
| NumTokSrc | -0.301 |
| PPL-SrcTgtIn | -0.631 |

Table 7: The feature importance coefficient for each feature, where a higher coefficient value implies greater feature importance.

## 5.2 Ablation studies

**Reference-less metric for learning regression model.** In the discussion above, the CTQ model was learnt to predict a reference-based metric (COMET). We also explored if a good CTQ metric can be learnt using a reference-less metric like COMET-QE (CTQ-QE). The results show that CTQ-QE outperforms the BM-25 baselines and is comparable to CTQ in many cases. Hence, the regression model can be effectively learnt even if a held-out parallel corpus is not available.

**Comparison of regression with averaging.** We also compared with a baseline (ScAvg) where the prompt was selected based on average scores of all features. In this ablation study, we considered

only 3 features *viz.* LaBSE-InSrc, LaBSE-InTgt, LaBSE-SrcTgt. We see that ScAvg already outperforms the corresponding individual features for most language pairs, hinting that even a simple combination of important features can be useful. We also see that the CTQ (3-feat) improves upon ScAvg (3-feat), showing that a regression-based example-based framework is important to elicit maximum translation quality.

**Neural *vs.* Linear Regression.** We compared using neural regression with linear regression on a few language pairs. The results are shown in Table 5. We observe that neural regression outperforms linear regression, hence justifying the choice of neural regression for example selection.

**Example Order.** We compared the ordering of examples from worst to best and vice-versa as per the CTQ score. Worst to best ordering seems to be better on an average (Table 6).

**Comparison of BM25 and Random Selection.** The example database consists of some parallel sentences with just 2 or 3 tokens. If these tokens are present in the input sample, the example is selected as a candidate using BM25 selection, though it is of low-relevance to the input. On the other hand, random selection might result in a reasonably good example being selected. In such cases, random selection outperforms BM25 selection. The learning is that such example pairs should be filtered from the database prior to selection.

**Analysis of feature salience.** To understand the importance of various features, we used a linear regression model to learn the CTQScorer for hi-en translation. The feature weights are interpretable and indicate feature importance. We observe that translation and example quality features are amongst the most important features. On the other hand, we observe that PPL-SrcTgtIn, NumTokSrc, and NumTokTgt are the least important features. The feature importance coefficients for the same are presented in Table 7.

## 6 Conclusion and Future Work

In this work, we propose an example selection method (CTQ Scorer) that utilizes multiple features for few-shot machine translation with LLMs. We show that combining multiple sentence-level features to predict the quality of retrieved examples results in significant improvement in translation quality over single features predicting example quality. Based on our ablation experiments, we also provide insights into the relevance of various features for example selection. Particularly, we would like to highlight that COMET-QE based similarity between input source and target is the best feature and is a strong baseline across languages and directions. We also note that the CTQScorer metric can be learnt without the need for a held-out seed parallel corpus. Exploration of example selection with multiple features for other NLP tasks is an interesting direction for future research.

## Limitations

In this paper, the BLOOM and XGLM models (both are multilingual language models), were primarily investigated. However, the generalizability of the findings to other language models is uncertain. The 7.1B and 7.5B models were used in our experiments, but it is possible that better results could be obtained with the larger models (like 176B model). Due to limitations in resources, our experimentation was restricted to only a few language pairs, and as such, our conclusions may differ if we had conducted experiments on a greater number of language pairs.

The proposed approach incurs inference time overhead for computing some features. However, the features can be computed in parallel and some features (like example quality) can be computed once offline - hence at least the latency can be controlled. There might be some scenarios like the need for high-quality domain-specific translation or generating training data where quality takes precedence over translation latency/cost. We could also limit ourselves to training a scorer model on a limited set of top-k features. We demonstrate the benefits of considering multiple factors in example selection, and we hope that in future, better methods can reduce the cost of feature selection.

## Ethics Statement

This work does not involve any new data collection and does not employ any annotators for data collection. We utilize publicly available datasets for the experiments reported in this work. Some of these datasets originate from web crawls, and we do not explicitly attempt to identify any biases within these datasets, using them in their original form.

## Acknowledgements

We would like to thank the Ministry of Electronics and Information Technology of the Government of India for their generous grant through the Digital India Bhashini project. We also thank the Centre for Development of Advanced Computing for providing compute time on the Param Siddhi Supercomputer. We also thank Nilekani Philanthropies for their generous grant towards building datasets, models, tools and resources for Indic languages. We also thank Microsoft for their grant to support research on Indic languages.

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

# A  Hyperparameter Details

We examined a diverse set of hyperparameters within specified ranges to tune our CTQ Scorer's neural regression model to achieve optimal results. Table 8 contains the range of explored hyperparameters. We also present the optimal hyperparameters employed for training each language pair's regression model in Table 10.

| Hyperparameter | Value |
|---|---|
| Number of Hidden layers | 3, 4, 5 |
| Neurons in Hidden Layer | 64, 128, 256, 512 |
| Activation Function | Sigmoid, Tanh, Relu |
| Batch Size | 16, 32, 64 |
| Learning Rate | 0.005, 0.001, 0.01 |
| Number of Epochs | 20, 30, 40 |
| Optimizer Algorithm | Adam, RMSprop, SGD |
| Weight Decay | 0, 0.005, 0.001, 0.01 |

Table 8: The range of hyperparameters used for finding the optimal hyperparameters, which are used for training the CTQ Neural Regression models.

| Selection Method | gu |
|---|---|
| Random Selection | -15.17 |
| BM25 | -9.17 |
| R-BM25 | -9.49 |
| CTQ (*ours*) | **4.15** |
| CTQ-QE (*ours*) | -1.11 |
| *Individual Features* | |
| NumTokSrc | -8.77 |
| NumTokTgt | -7.05 |
| CmtQE-InSrc | -7.83 |
| CmtQE-InTgt | 3.51 |
| CmtQE-SrcTgt | -3.06 |
| chrF-InSrc | -10.34 |
| LaBSE-InSrc | -2.99 |
| LaBSE-InTgt | -3.59 |
| LaBSE-SrcTgt | -12.95 |
| PPL-SrcTgt | -4.05 |
| PPL-SrcTgtIn | 0.31 |
| *Comparison with Score Averaging* | |
| ScAvg (3-feat) | -1.71 |
| CTQ (3-feat) | -5.43 |

Table 9: COMET scores for translation out of English to Gujarati using different example selection methods. The highest scores are in **bold** text.

## B  Other Metric Results

This section presents the main example selection for COMET-22 (Rei et al., 2022) using *wmt22-comet-da* model (Tables 11, 12). We observe that the trends observed for COMET-20 also hold for COMET-22.

We also report COMET-QE (Rei et al., 2020b) using *wmt20-comet-qe-da* model (Tables 14, 13), BLEU (Tables 15, 16) (Post, 2018)[4], chrF (Tables 17, 18) (Popović, 2015)[5], and chrF++ (Tables 19, 20) (Popović, 2017)[6] for the sake of completeness.

## C  Off-target Translation

This section presents the off-target translation percentages when translating into and from English using different example selection methods (Tables

21, 22). We used the NLLB language identifier (LID) (NLLB Team et al., 2022) to identify the target language, considering the sentence to be in the target language if the LID score is greater than 0.9.

## D  Translation scores for en-gu

COMET *wmt20-comet-da* translation scores for en-gu are negative for any translation example pair. We think that's because COMET *wmt20-comet-da* isn't trained well on gu data. However, for completeness we added the translation scores for en-gu in Table 9.

---

[4]nrefs:1|case:mixed|eff:no|tok:13a|smooth:exp|version:2.3.1

[5]chrF2|nrefs:1|case:mixed|eff:yes|nc:6|nw:0|space:no|version:2.0.0

[6]chrF2++|nrefs:1|case:mixed|eff:yes|nc:6|nw:2|space:no|version:2.0.0

| MT of XX-XX | Number of Hidden layers | Neurons in Hidden Layer | Activation Function | Batch Size | Learning Rate | Number of Epochs | Optimizer Algorithm | Weight Decay |
|---|---|---|---|---|---|---|---|---|
| bn-en | 5 | 256 | Relu | 16 | 0.001 | 30 | Adam | 0 |
| de-en | 4 | 256 | Sigmoid | 16 | 0.005 | 40 | Adam | 0 |
| fr-en | 5 | 256 | Relu | 32 | 0.001 | 40 | RMSProp | 0 |
| gu-en | 4 | 512 | Relu | 32 | 0.001 | 20 | Adam | 0 |
| hi-en | 3 | 128 | Relu | 32 | 0.001 | 30 | RMSProp | 0 |
| ru-en | 5 | 256 | Relu | 32 | 0.001 | 20 | RMSProp | 0 |
| bn-en | 3 | 128 | Relu | 32 | 0.001 | 20 | Adam | 0 |
| en-de | 4 | 256 | Relu | 64 | 0.005 | 40 | RMSProp | 0 |
| en-fr | 4 | 128 | Tanh | 64 | 0.001 | 30 | RMSProp | 0 |
| en-gu | 4 | 512 | Relu | 16 | 0.001 | 40 | RMSProp | 0 |
| en-hi | 3 | 128 | Relu | 64 | 0.001 | 40 | Adam | 0 |
| en-ru | 3 | 128 | Tanh | 32 | 0.001 | 40 | Adam | 0 |

Table 10: Optimal hyperparameters that were employed for training the CTQ Neural Regression Model for each language pair.

| Selection Method | bn | gu | hi | de | fr | ru | Average |
|---|---|---|---|---|---|---|---|
| Random Selection | 82.61 | 81.99 | 83.44 | 86.36 | 86.58 | 82.88 | 83.98 |
| BM25 | 82.38 | 82.12 | 83.50 | 86.14 | 86.69 | 82.34 | 83.86 |
| R-BM25 | 82.50 | 81.97 | 83.51 | 86.34 | 86.66 | 82.95 | 83.99 |
| CTQ (*ours*) | 83.25 | 82.71 | 84.37 | **86.66** | 86.78 | 83.31 | **84.51** |
| *Individual Features* | | | | | | | |
| NumTokSrc | 82.36 | 82.39 | 82.73 | 86.19 | 86.57 | 82.66 | 83.82 |
| NumTokTgt | 82.61 | 81.31 | 82.13 | 86.08 | 86.61 | 82.32 | 83.51 |
| CmtQE-InSrc | 82.53 | 81.96 | 83.49 | 86.43 | 86.55 | 83.18 | 84.02 |
| CmtQE-InTgt | 82.79 | 82.46 | 84.06 | 86.61 | 86.53 | **83.34** | 84.30 |
| CmtQE-SrcTgt | 82.72 | 82.22 | 84.24 | 86.20 | 86.56 | 82.48 | 84.07 |
| chrF-InSrc | 82.18 | 81.76 | 83.23 | 85.62 | 86.41 | 80.82 | 83.34 |
| LaBSE-InSrc | 82.56 | 81.89 | 83.86 | 86.31 | 86.66 | 82.71 | 84.00 |
| LaBSE-InTgt | 82.94 | 82.10 | 84.03 | 86.12 | 86.64 | 81.96 | 83.97 |
| LaBSE-SrcTgt | 82.96 | 82.28 | 83.97 | 86.10 | 86.44 | 80.70 | 83.74 |
| PPL-SrcTgt | 82.80 | 82.85 | 83.44 | 86.38 | 86.71 | 82.59 | 84.13 |
| PPL-SrcTgtIn | 82.70 | **82.78** | 83.64 | 86.37 | **86.85** | 82.54 | 84.15 |
| *Comparison with Score Averaging* | | | | | | | |
| ScAvg (3-feat) | **83.28** | 82.66 | 84.17 | 86.19 | 86.65 | 81.82 | 84.13 |
| CTQ (3-feat) | 83.11 | 82.44 | **84.39** | 86.42 | 86.73 | 82.93 | 84.34 |

Table 11: COMET-22 scores for translation into English using different example selection methods. The highest scores are in **bold** text.

| Selection Method | bn | gu | hi | de | fr | ru | Average |
|---|---|---|---|---|---|---|---|
| Random Selection | 77.32 | 66.30 | 70.44 | 79.41 | 80.22 | 80.11 | 75.63 |
| BM25 | 77.69 | 67.34 | 70.48 | 79.85 | 80.24 | 81.45 | 76.18 |
| R-BM25 | 77.92 | 67.51 | 70.91 | 79.90 | 80.23 | 82.00 | 76.41 |
| CTQ (*ours*) | 78.56 | **71.21** | 71.20 | 80.22 | 80.37 | 83.10 | 77.44 |
| *Individual Features* | | | | | | | |
| NumTokSrc | **79.37** | 67.88 | 70.21 | 78.71 | 79.69 | 80.90 | 76.13 |
| NumTokTgt | 79.18 | 68.71 | 70.22 | 79.04 | 79.38 | 81.12 | 76.28 |
| CmtQE-InSrc | 78.23 | 68.34 | 70.91 | 80.28 | 80.60 | 82.19 | 76.76 |
| CmtQE-InTgt | 78.48 | 70.67 | **71.77** | **80.30** | **80.89** | **83.27** | **77.56** |
| CmtQE-SrcTgt | 78.57 | 69.44 | 70.86 | 79.66 | 80.14 | 82.73 | 76.90 |
| chrF-InSrc | 77.77 | 67.32 | 70.46 | 79.50 | 79.80 | 81.40 | 76.04 |
| LaBSE-InSrc | 77.95 | 69.10 | 70.52 | 79.42 | 80.32 | 81.96 | 76.55 |
| LaBSE-InTgt | 78.80 | 69.36 | 71.47 | 79.83 | 80.10 | 82.23 | 76.97 |
| LaBSE-SrcTgt | 77.62 | 66.86 | 69.60 | 79.17 | 79.17 | 80.13 | 75.43 |
| PPL-SrcTgt | 79.26 | 69.10 | 71.13 | 79.85 | 80.36 | 81.54 | 76.87 |
| PPL-SrcTgtIn | 78.89 | 70.30 | 71.33 | 79.08 | 79.57 | 81.59 | 76.79 |
| *Comparison with Score Averaging* | | | | | | | |
| ScAvg (3-feat) | 78.38 | 69.53 | 71.74 | 79.88 | 80.31 | 81.78 | 76.94 |
| CTQ (3-feat) | 78.56 | 68.97 | 71.66 | 79.84 | 80.41 | 82.43 | 76.98 |

Table 12: COMET-22 scores for translation out of English using different example selection methods. The highest scores are in **bold** text.

| Selection Method | bn | gu | hi | de | fr | ru | Average |
|---|---|---|---|---|---|---|---|
| Random Selection | 35.4 | 36.42 | 39.44 | 46.16 | 33.02 | 33.47 | 37.32 |
| BM25 | 35.41 | 36.88 | 39.51 | 44.91 | 32.51 | 31.36 | 36.76 |
| R-BM25 | 36.21 | 37.45 | 39.75 | 46.27 | 33.02 | 33.64 | 37.72 |
| CTQ (*ours*) | 36.96 | 38.56 | 39.59 | **46.92** | 33.45 | 34.86 | 38.39 |
| CTQ-QE (*ours*) | 36.17 | 39.35 | 40.66 | 46.02 | **33.52** | 34.81 | 38.42 |
| *Individual Features* | | | | | | | |
| NumTokSrc | 36.53 | 38.6 | 37.34 | 45.38 | 32.83 | 32.57 | 37.21 |
| NumTokTgt | 37.38 | 38.55 | 38.06 | 45.65 | 32.85 | 32.22 | 37.45 |
| CmtQE-InSrc | 37.56 | 38.55 | 40.98 | 46.77 | 32.42 | 34.86 | 38.52 |
| CmtQE-InTgt | **38.48** | **40.02** | **42.09** | 46.89 | 32.63 | **35.35** | **39.24** |
| CmtQE-SrcTgt | 36.74 | 37.68 | 41.16 | 45.24 | 32.04 | 31.79 | 37.44 |
| chrF-InSrc | 35.32 | 36.83 | 39.22 | 42.48 | 31.71 | 25.26 | 35.14 |
| LaBSE-InSrc | 36.81 | 37.44 | 39.88 | 45.68 | 32.55 | 32.38 | 37.46 |
| LaBSE-InTgt | 37.14 | 37.85 | 40.41 | 45.03 | 32.47 | 29.64 | 37.09 |
| LaBSE-InTgt | 35.68 | 36.59 | 38.57 | 44.21 | 32.12 | 24.45 | 35.27 |
| PPL-SrcTgt | 36.69 | 39.37 | 40.73 | 45.98 | 33.13 | 31.91 | 37.97 |
| PPL-SrcTgtIn | 36.82 | 39.26 | 39.32 | 45.54 | 33.39 | 31.76 | 37.68 |

Table 13: COMET-QE-20 scores for translation into English using different example selection methods. The highest scores are in **bold** text.

| Selection Method | bn | gu | hi | de | fr | ru | Average |
|---|---|---|---|---|---|---|---|
| Random Selection | 25.59 | 3.3 | 17.55 | 30.9 | 20 | 23.09 | 20.07 |
| BM25 | 24.48 | 6.59 | 16.71 | 30.78 | 19.67 | 27.49 | 20.95 |
| R-BM25 | 24.99 | 8.37 | 19.19 | 32.22 | 18.93 | 29.07 | 22.13 |
| CTQ (*ours*) | 30.06 | 21.86 | 21.95 | 33.22 | 18.76 | 33.62 | 26.58 |
| CTQ-QE (*ours*) | 29.09 | 18.25 | 22.8 | 31.55 | 19.19 | 32.87 | 25.63 |
| *Individual Features* | | | | | | | |
| NumTokSrc | 28.29 | 9.97 | 15.1 | 27.27 | 15.25 | 26.3 | 20.36 |
| NumTokTgt | 27.85 | 11.71 | 15.76 | 27.86 | 15.07 | 26.31 | 20.76 |
| CmtQE-InSrc | 27.61 | 11.51 | 19.64 | 32.68 | 19.86 | 30.52 | 23.64 |
| CmtQE-InTgt | **30.85** | **22.25** | **24.13** | **34.38** | **21.07** | **34.64** | **27.89** |
| CmtQE-SrcTgt | 30.72 | 16.92 | 21.01 | 32.52 | 19.47 | 33.06 | 25.62 |
| chrF-InSrc | 25.28 | 8.22 | 18.57 | 30.88 | 17.33 | 27.27 | 21.26 |
| LaBSE-InSrc | 26.42 | 15.42 | 18.1 | 30.48 | 19.12 | 28.87 | 23.07 |
| LaBSE-InTgt | 27.87 | 13.61 | 19.22 | 31.59 | 18.48 | 29.82 | 23.43 |
| LaBSE-InTgt | 22.87 | 4.29 | 12.8 | 30.36 | 14.61 | 22.93 | 17.98 |
| PPL-SrcTgt | 28.26 | 13.94 | 18.55 | 31.21 | 18.88 | 28.19 | 23.17 |
| PPL-SrcTgtIn | 26.69 | 17.77 | 19.13 | 27.93 | 15.98 | 27.65 | 22.53 |

Table 14: COMET-QE-20 scores for translation out of English using different example selection methods. The highest scores are in **bold** text.

| Selection Method | bn | gu | hi | de | fr | ru | Average |
|---|---|---|---|---|---|---|---|
| Random Selection | 18.60 | 17.18 | 20.73 | 33.91 | 35.17 | 26.66 | 25.38 |
| BM25 | 19.43 | 17.08 | 22.18 | 33.96 | **35.75** | 25.96 | 25.73 |
| R-BM25 | 19.11 | 16.91 | 21.52 | 33.99 | 35.38 | 26.37 | 25.55 |
| CTQ (*ours*) | 20.44 | **18.26** | 23.00 | **34.92** | 35.64 | **27.34** | **26.60** |
| *Individual Features* | | | | | | | |
| CmtQE-InSrc | 18.48 | 16.44 | 20.78 | 34.56 | 35.69 | 26.73 | 25.45 |
| CmtQE-InTgt | 18.85 | 16.95 | 21.71 | 34.91 | 35.67 | 26.91 | 25.83 |
| CmtQE-SrcTgt | 18.62 | 17.29 | 21.71 | 34.89 | 35.29 | 26.60 | 25.73 |
| chrF-InSrc | 18.82 | 16.33 | 22.09 | 33.28 | 35.27 | 24.69 | 25.08 |
| LaBSE-InSrc | 19.54 | 16.64 | 22.79 | 34.28 | 35.52 | 26.41 | 25.86 |
| LaBSE-InTgt | 19.85 | 16.83 | 22.70 | 33.97 | 35.02 | 25.53 | 25.65 |
| LaBSE-SrcTgt | 18.99 | 17.47 | 21.89 | 34.02 | 34.97 | 24.66 | 25.33 |
| PPL-SrcTgt | 19.42 | 17.32 | 21.28 | 34.46 | 35.34 | 26.70 | 25.75 |
| PPL-SrcTgtIn | 19.64 | 17.40 | 22.57 | 34.55 | 35.43 | 26.93 | 26.09 |
| *Comparison with Score Averaging* | | | | | | | |
| ScAvg (3-feat) | **20.70** | 17.81 | 22.62 | 34.17 | 35.57 | 25.54 | 26.07 |
| CTQ (3-feat) | 20.24 | 17.56 | **23.46** | 34.58 | 35.73 | 26.79 | 26.39 |

Table 15: BLEU scores for translation into English using different example selection methods.. The highest scores are in **bold** text.

| Selection Method | bn | gu | hi | de | fr | ru | Average |
|---|---|---|---|---|---|---|---|
| Random Selection | 5.25 | 4.61 | 13.64 | 20.48 | 29.13 | 17.99 | 15.18 |
| BM25 | 5.93 | 4.91 | 13.89 | 21.16 | 29.79 | 18.43 | 15.69 |
| R-BM25 | 5.85 | 4.57 | 14.18 | 21.07 | 29.57 | 18.54 | 15.63 |
| CTQ (*ours*) | 5.77 | 5.07 | 14.18 | **21.86** | 29.58 | 18.69 | 15.86 |
| *Individual Features* | | | | | | | |
| CmtQE-InSrc | 5.74 | 4.48 | 14.02 | 21.07 | 30.62 | 18.27 | 15.70 |
| CmtQE-InTgt | 5.40 | 4.35 | 13.35 | 21.38 | 30.46 | 17.94 | 15.48 |
| CmtQE-SrcTgt | 5.41 | 4.75 | 13.46 | 20.80 | 29.75 | 17.85 | 15.34 |
| chrF-InSrc | 6.02 | 4.52 | 13.73 | 21.08 | 29.90 | 18.08 | 15.56 |
| LaBSE-InSrc | 5.96 | 4.65 | 14.02 | 20.83 | 30.48 | 18.76 | 15.78 |
| LaBSE-InTgt | 6.03 | 4.93 | 14.64 | 20.88 | 29.85 | 19.01 | 15.89 |
| LaBSE-SrcTgt | **6.78** | 4.99 | 15.16 | 21.60 | 29.22 | 18.14 | 15.98 |
| PPL-SrcTgt | **6.78** | 5.27 | 15.14 | 21.61 | **30.76** | 18.13 | **16.28** |
| PPL-SrcTgtIn | 6.52 | 5.10 | **15.55** | 21.22 | 29.64 | 18.51 | 16.09 |
| *Comparison with Score Averaging* | | | | | | | |
| ScAvg (3-feat) | 6.59 | **5.29** | 15.24 | 21.21 | 30.34 | 18.43 | 16.18 |
| CTQ (3-feat) | 6.27 | 4.90 | 15.32 | 21.64 | 30.40 | **19.13** | **16.28** |

Table 16: BLEU scores for translation out of English using different example selection methods. The highest scores are in **bold** text.

| Selection Method | bn | gu | hi | de | fr | ru | Average |
|---|---|---|---|---|---|---|---|
| Random Selection | 46.10 | 44.37 | 48.70 | 60.52 | 61.59 | 55.08 | 52.73 |
| BM25 | 46.39 | 44.04 | 49.69 | 60.37 | 61.89 | 54.30 | 52.78 |
| R-BM25 | 46.66 | 44.09 | 49.15 | 60.18 | 61.77 | 54.97 | 52.80 |
| CTQ (*ours*) | 47.79 | 45.25 | **51.44** | **61.19** | 61.92 | **55.60** | **53.87** |
| *Individual Features* | | | | | | | |
| CmtQE-InSrc | 45.54 | 43.43 | 48.56 | 60.76 | 61.72 | 55.03 | 52.51 |
| CmtQE-InTgt | 45.89 | 44.02 | 49.55 | 61.13 | 61.83 | 55.15 | 52.93 |
| CmtQE-SrcTgt | 46.09 | 44.78 | 49.85 | 60.95 | 61.79 | 54.64 | 53.02 |
| chrF-InSrc | 46.11 | 43.62 | 48.96 | 59.86 | 61.69 | 53.67 | 52.32 |
| LaBSE-InSrc | 46.36 | 43.39 | 50.13 | 60.76 | 62.00 | 55.02 | 52.94 |
| LaBSE-InTgt | 46.66 | 43.65 | 50.17 | 60.44 | 61.84 | 54.29 | 52.84 |
| LaBSE-SrcTgt | 46.91 | **45.59** | 50.50 | 60.89 | 61.88 | 53.29 | 53.18 |
| PPL-SrcTgt | 46.95 | 44.93 | 49.44 | 60.86 | 61.98 | 54.89 | 53.18 |
| PPL-SrcTgtIn | 46.92 | 45.04 | 50.17 | 61.01 | 62.05 | 54.89 | 53.35 |
| *Comparison with Score Averaging* | | | | | | | |
| ScAvg (3-feat) | 47.67 | 45.14 | 50.15 | 60.90 | 62.02 | 54.39 | 53.38 |
| CTQ (3-feat) | **47.94** | 44.71 | 51.42 | 60.93 | **62.20** | 55.54 | 53.79 |

Table 17: chrF scores for translation into English using different example selection methods.. The highest scores are in **bold** text.

| Selection Method | bn | gu | hi | de | fr | ru | Average |
|---|---|---|---|---|---|---|---|
| Random Selection | 32.05 | 23.13 | 37.23 | 50.50 | 55.06 | 46.17 | 40.69 |
| BM25 | 34.20 | 24.10 | 38.33 | 51.17 | 55.59 | 46.61 | 41.67 |
| R-BM25 | 34.18 | 23.61 | 38.33 | 51.08 | 55.50 | 46.94 | 41.61 |
| CTQ (*ours*) | 33.68 | 25.22 | 38.33 | 51.69 | 55.42 | 47.18 | 41.92 |
| *Individual Features* | | | | | | | |
| CmtQE-InSrc | 34.06 | 23.87 | 38.05 | 51.22 | 56.28 | 46.53 | 41.67 |
| CmtQE-InTgt | 33.13 | 24.49 | 37.82 | 51.17 | 56.06 | 46.34 | 41.50 |
| CmtQE-SrcTgt | 33.08 | 24.29 | 37.26 | 50.75 | 55.25 | 46.25 | 41.15 |
| chrF-InSrc | 33.96 | 23.75 | 37.91 | 50.87 | 55.27 | 46.43 | 41.37 |
| LaBSE-InSrc | 33.81 | 24.27 | 37.96 | 51.15 | 55.89 | 46.83 | 41.65 |
| LaBSE-InTgt | 34.68 | 24.67 | 38.98 | 51.10 | 55.56 | 47.24 | 42.04 |
| LaBSE-SrcTgt | 35.05 | 24.43 | 39.33 | 51.26 | 54.88 | 45.97 | 41.82 |
| PPL-SrcTgt | **35.78** | **25.24** | 39.46 | 51.59 | **56.47** | 46.21 | 42.46 |
| PPL-SrcTgtIn | 35.66 | 25.51 | 39.91 | 51.18 | 55.4 | 46.72 | 42.40 |
| *Comparison with Score Averaging* | | | | | | | |
| ScAvg (3-feat) | 35.37 | 25.10 | 39.95 | 51.32 | 55.84 | 47.15 | 42.46 |
| CTQ (3-feat) | 35.23 | 24.63 | **40.05** | **51.78** | 55.95 | **47.47** | **42.52** |

Table 18: chrF scores for translation out of English using different example selection methods. The highest scores are in **bold** text.

| Selection Method | bn | gu | hi | de | fr | ru | Average |
|---|---|---|---|---|---|---|---|
| Random Selection | 43.80 | 42.12 | 46.28 | 58.63 | 59.82 | 52.87 | 50.59 |
| BM25 | 44.11 | 41.82 | 47.29 | 58.47 | 60.10 | 52.07 | 50.64 |
| R-BM25 | 44.29 | 41.84 | 46.80 | 58.29 | 60.01 | 52.74 | 50.66 |
| CTQ (*ours*) | 45.42 | 43.01 | **49.03** | **59.35** | 60.18 | **53.41** | **51.73** |
| *Individual Features* | | | | | | | |
| CmtQE-InSrc | 43.20 | 41.16 | 46.17 | 58.93 | 59.98 | 52.86 | 50.38 |
| CmtQE-InTgt | 43.53 | 41.74 | 47.20 | 59.27 | 60.04 | 52.97 | 50.79 |
| CmtQE-SrcTgt | 43.81 | 42.54 | 47.49 | 59.10 | 60.00 | 52.42 | 50.89 |
| chrF-InSrc | 43.78 | 41.43 | 46.67 | 57.97 | 59.93 | 51.43 | 50.20 |
| LaBSE-InSrc | 44.05 | 41.18 | 47.75 | 58.89 | 60.19 | 52.78 | 50.81 |
| LaBSE-InTgt | 44.34 | 41.51 | 47.81 | 58.52 | 60.05 | 52.03 | 50.71 |
| LaBSE-SrcTgt | 44.57 | **43.26** | 48.05 | 58.98 | 60.06 | 51.02 | 50.99 |
| PPL-SrcTgt | 44.55 | 42.62 | 47.00 | 58.98 | 60.22 | 52.64 | 51.00 |
| PPL-SrcTgtIn | 44.51 | 42.71 | 47.83 | 59.10 | 60.25 | 52.65 | 51.18 |
| *Comparison with Score Averaging* | | | | | | | |
| ScAvg (3-feat) | 45.34 | 42.91 | 47.83 | 59.03 | 60.23 | 52.13 | 51.25 |
| CTQ (3-feat) | **45.55** | 42.46 | **49.03** | 59.06 | **60.42** | 53.34 | 51.64 |

Table 19: chrF++ scores for translation into English using different example selection methods.. The highest scores are in **bold** text.

| Selection Method | bn | gu | hi | de | fr | ru | Average |
|---|---|---|---|---|---|---|---|
| Random Selection | 28.45 | 21.46 | 35.51 | 47.40 | 52.65 | 43.14 | 38.10 |
| BM25 | 30.39 | 22.29 | 36.44 | 48.07 | 53.21 | 43.53 | 38.99 |
| R-BM25 | 30.34 | 21.80 | 36.50 | 47.97 | 53.10 | 43.83 | 38.92 |
| CTQ (*ours*) | 29.85 | 23.38 | 36.45 | **48.66** | 52.99 | 44.04 | 39.23 |
| *Individual Features* | | | | | | | |
| CmtQE-InSrc | 30.21 | 21.98 | 36.19 | 48.15 | 53.89 | 43.41 | 38.97 |
| CmtQE-InTgt | 29.29 | 22.56 | 35.89 | 48.12 | 53.66 | 43.23 | 38.79 |
| CmtQE-SrcTgt | 29.28 | 22.48 | 35.51 | 47.68 | 52.90 | 43.13 | 38.50 |
| chrF-InSrc | 30.14 | 21.95 | 36.04 | 47.82 | 52.89 | 43.33 | 38.70 |
| LaBSE-InSrc | 29.94 | 22.40 | 36.11 | 48.06 | 53.55 | 43.76 | 38.97 |
| LaBSE-InTgt | 30.79 | 22.80 | 37.10 | 48.01 | 53.15 | 44.13 | 39.33 |
| LaBSE-SrcTgt | 31.33 | 22.71 | 37.52 | 48.17 | 52.46 | 42.90 | 39.18 |
| PPL-SrcTgt | **31.86** | 23.32 | 37.65 | 48.57 | **54.07** | 43.10 | 39.76 |
| PPL-SrcTgtIn | 31.74 | **23.59** | 38.07 | 48.08 | 53.00 | 43.56 | 39.67 |
| *Comparison with Score Averaging* | | | | | | | |
| ScAvg (3-feat) | 31.49 | 23.27 | 38.07 | 48.22 | 53.48 | 43.97 | 39.75 |
| CTQ (3-feat) | 31.32 | 22.76 | **38.18** | 48.65 | 53.55 | **44.36** | **39.80** |

Table 20: chrF++ scores for translation out of English using different example selection methods. The highest scores are in **bold** text.

| Selection Method | bn | gu | hi | de | fr | ru | Average |
|---|---|---|---|---|---|---|---|
| Random Selection | 1.28 | 0.59 | 2.27 | 0.30 | 0.10 | 0.49 | 0.84 |
| BM25 | 1.38 | 0.59 | 1.48 | 0.30 | 0.40 | 1.78 | 0.99 |
| R-BM25 | 0.99 | 0.89 | 2.27 | 0.10 | 0.30 | 0.69 | 0.87 |
| CTQ (*ours*) | **0.79** | **0.40** | **0.69** | 0.10 | 0.20 | 0.40 | **0.43** |
| CTQ-QE (*ours*) | 1.28 | 0.69 | 1.98 | 0.10 | 0.20 | **0.20** | 0.74 |
| *Individual Features* | | | | | | | |
| NumTokSrc | 2.47 | 0.69 | 3.56 | 0.10 | **0.10** | 0.49 | 1.24 |
| NumTokTgt | 1.58 | 0.89 | 3.66 | 0.20 | **0.10** | 0.30 | 1.12 |
| CmtQE-InSrc | **0.79** | 0.49 | 2.37 | 0.00 | 0.40 | 0.49 | 0.76 |
| CmtQE-InTgt | 1.19 | **0.40** | 1.98 | 0.00 | 0.49 | 0.20 | 0.71 |
| CmtQE-SrcTgt | 0.99 | 0.79 | 1.09 | 0.49 | 0.79 | 1.68 | 0.97 |
| chrF-InSrc | 1.28 | 0.59 | 1.78 | 1.48 | 0.79 | 4.35 | 1.71 |
| LaBSE (Query - Src) | 1.09 | 0.40 | 1.78 | 0.20 | 0.40 | 1.48 | 0.89 |
| LaBSE-InTgt | 1.19 | 0.59 | 1.28 | 0.59 | 0.40 | 2.17 | 1.04 |
| LaBSE-SrcTgt | 1.09 | 1.19 | 1.68 | 0.79 | 0.69 | 4.05 | 1.58 |
| PPL-SrcTgt | 1.09 | 1.28 | 2.77 | **0.00** | 0.20 | 0.69 | 1.01 |
| PPL-SrcTgtIn | 1.78 | 0.89 | 2.57 | 0.20 | 0.20 | 0.59 | 1.04 |
| *Comparison with Score Averaging* | | | | | | | |
| ScAvg (3-feat) | 0.99 | 0.59 | 1.58 | 0.69 | 0.49 | 2.37 | 1.12 |
| CTQ (3-feat) | 0.89 | 0.59 | 1.38 | 0.10 | 0.40 | 0.99 | 0.72 |

Table 21: Off-target percentage for translation into English using different example selection methods. The lowest scores are in **bold** text.

| Selection Method | bn | gu | hi | de | fr | ru | Average |
|---|---|---|---|---|---|---|---|
| Random Selection | 2.27 | 6.62 | 4.84 | 1.09 | **0.89** | 1.09 | 2.80 |
| BM25 | 1.98 | 4.94 | 4.15 | 1.28 | 1.58 | 1.78 | 2.62 |
| R-BM25 | 1.58 | 5.14 | 3.06 | 0.59 | 1.38 | 1.28 | 2.17 |
| CTQ (*ours*) | 1.48 | 3.85 | 3.36 | 0.59 | 1.28 | **0.79** | 1.89 |
| CTQ-QE (*ours*) | **0.99** | 4.64 | **2.57** | 0.69 | 1.28 | 1.09 | 1.88 |
| *Individual Features* | | | | | | | |
| NumTokSrc | **0.99** | 4.35 | 4.74 | 0.79 | 0.99 | **0.79** | 2.11 |
| NumTokTgt | 1.38 | 5.83 | 4.35 | **0.49** | 1.09 | 1.28 | 2.40 |
| CmtQE-InSrc | 1.28 | 5.24 | 3.36 | 0.99 | 1.68 | 1.09 | 2.27 |
| CmtQE-InTgt | 1.09 | **3.75** | 2.47 | 1.09 | 1.28 | 0.99 | **1.78** |
| CmtQE-SrcTgt | 1.19 | 4.45 | 4.05 | 0.89 | 1.58 | 1.58 | 2.29 |
| chrF-InSrc | 1.58 | 5.43 | 2.87 | 1.38 | 1.78 | 2.27 | 2.55 |
| LaBSE (Query - Src) | 1.48 | 3.95 | 2.77 | 1.19 | 1.68 | 1.68 | 2.13 |
| LaBSE-InTgt | 1.48 | 4.45 | 3.26 | 1.19 | 1.48 | 1.48 | 2.22 |
| LaBSE-SrcTgt | 2.96 | 6.42 | 4.25 | 2.37 | 3.16 | 3.95 | 3.85 |
| PPL-SrcTgt | 1.68 | 5.14 | 3.85 | 0.59 | 0.99 | 1.38 | 2.27 |
| PPL-SrcTgtIn | 2.08 | **3.75** | 3.66 | 0.69 | **0.89** | 1.28 | 2.06 |
| *Comparison with Score Averaging* | | | | | | | |
| ScAvg (3-feat) | 1.98 | 4.45 | 3.16 | 1.19 | 1.98 | 1.38 | 2.36 |
| CTQ (3-feat) | 1.38 | 5.63 | 3.75 | 0.79 | 1.78 | 1.09 | 2.40 |

Table 22: Off-target percentage for translation from English using different example selection methods. The lowest scores are in **bold** text.