# OpenReview forum: "CTQScorer: Combining Multiple Features for In-context Example Selection for Machine Translation"
_EMNLP/2023/Conference — EMNLP 2023 Findings_

### Official Review · Reviewer_Tf25 · 2023-07-25

**Soundness:** 3

**Excitement:**

4: Strong: This paper deepens the understanding of some phenomenon or lowers the barriers to an existing research direction.

**Paper Topic And Main Contributions:**

The draft proposes a learnable scorer to select example for in-context learning in MT. The scorer is a shallow MLP regressor with COMET targets on 11 tabular features, extracted from triples of (in-context input $x_p$, target $y_p$, an example's input $x$).

The method delivers significant gains in COMET for low and hi-resource language on two datasets.

To my point of view, the biggest discovery of this paper is that you don't need a complicated scorer to do the selection, just selecting examples by COMET$(y_p, x)$ is enough to reap almost all gains from tuning the in-context prompt, and is much simpler and beats all the baselines on its own. I'd recommend highlighting this as the main finding in the paper.

**Questions For The Authors:**

- Since the linear regressor was also trained could you select the 3 most salient feature instead of LABSE- features in the the comparison to averaging?
- In Table 4, for fr and ru, CmtQE-InTgt beats all methods, including the proposed CTQScorer. Why was the scorer unable to learn to ignore the rest of features and just use this feature?
- Table 4: could you add the en-gu column as well? It's confusing that the en-gu column in Table 5 cannot be verified in Table 4 unlike other languages.
- Table 6: is it from or into English?



**Reasons To Accept:**

- significant gains in translation quality
- comprehensive analysis and ablations

**Reasons To Reject:**

- more complicated model is proposed than just using one of the 11 features, which is almost as good as using the full MLP (the focus of the paper can be shifted to this finding)
- lack of analysis of feature salience, which could be added for example to the "Neural vs. Linear Regression" experiments.


**Reproducibility:**

3: Could reproduce the results with some difficulty. The settings of parameters are underspecified or subjectively determined; the training/evaluation data are not widely available.

**Reviewer Confidence:**

4: Quite sure. I tried to check the important points carefully. It's unlikely, though conceivable, that I missed something that should affect my ratings.

**Typos Grammar Style And Presentation Improvements:**

- Please mention which sentence-level MT metric for regressor you are actually using to train the scorer as early as possible, e.g. in 3.3. As far as I could tell, it's only made explicit in 5.2 (line 532)

---

> ### Author Rebuttal · Authors · 2023-08-29
>
> Thank you for your time and helpful comments! We address your concerns below:
>
> **-Q1: Highlight COMET$(y_p, x)$ is enough to reap almost all gains from tuning the in-context prompt**
>
> Thanks for the feedback. We have mentioned it in the introduction section of the paper  and we will highlight it further.
>
> **-Q2: More complicated model is proposed than just using one of the 11 features**
>
> While our model is indeed complex, the consistent gains more than compensate for it. We will utilise the additional space in the camera-ready version to focus on analyses of individual features in comparison with the combination approach. Moreover, it's worth noting that when translating to English, our CTQ model outperforms the CmtQE-InTgt approach.
>
> **-Q3: Lack of analysis of feature salience, which could be added for example to the "Neural vs. Linear Regression" experiments**
>
> We did the analysis for linear regression but could not include it due to space constraints. We observed that PPL-SrcTgt, LaBSE-InSrc, LaBSE-SrcTgt, and Cmt-InTgt are the most important features. On the other hand, we found NumTokIn, NumTokSrc, and NumTokTgt to be the least important features. We will include the same findings in the camera-ready version.
>
> **-Q4: Could you select the 3 most salient feature instead of LABSE-features in the the comparison to averaging**
>
> That is an interesting idea to try. When doing this experiment, our motivation was to establish the need for regression over simple averaging.
>
> **-Q5: In Table 4, for fr and ru, CmtQE-InTgt beats all methods, including the proposed CTQScorer. Why was the scorer unable to learn to ignore the rest of features and just use this feature?**
>
> For the en-ru pair, the performance between CTQScorer and CmtQE-InTgt is quite competitive, with marginal differences.  However, the discrepancy is more pronounced for en-fr. Theoretically, if one feature (like CmtQE-InTgt) is overwhelmingly superior for a specific language pair, our model should be able to recognize and prioritize it. However, due to the complex interplay of features, CTQScorer might sometimes spread its emphasis across multiple features, rather than singularly focusing on the most effective one.
>
> **-Q6: Table 4: Could you add the en-gu column as well?**
>
> COMET wmt20-comet-da translation scores for en-gu are negative for any translation example pair. We think that’s because COMET wmt20-comet-da isn’t trained well on gu data. That is why we didn’t include the en-gu column in table 5. We will add the same to the paper for completeness.
>
> **-Q7: Table 6: is it from or into English?**
>
> It is into English. We will update the caption accordingly.
>
> **-Q8: Typos: Mention which sentence-level MT metric for regressor you are actually using to train the scorer as early as possible**
>
> Thanks for the feedback. We will update in the camera-ready version.

---

### Official Review · Reviewer_STxx · 2023-08-05

**Soundness:** 4

**Excitement:**

4: Strong: This paper deepens the understanding of some phenomenon or lowers the barriers to an existing research direction.

**Paper Topic And Main Contributions:**

The paper proposes a new CTQ Scorer model that utilizes multiple features for few-shot machine translation with LLMs. The model seems to do well on selecting examples based on a diverse set of features. The features are computed regarding regarding the relevancy of translation examples for an input, and also the translation quality of the translation examples themselves. The features are also computed by using various metrics such as LaBSE or COMET-QE. Experiments show that combining multiple sentence-level features with CTQ scorer model to predict the quality of retrieved examples results in significant improvement in translation quality over single features predicting example quality.

**Questions For The Authors:**

I am not quite clear yet about the novelty of the work. My reviews are based on the assumption that the work is novel regarding having a model (i.e CTQ scorer) that combines different features to learn how to score the relevancy of a translation example to an input. But I don't understand yet the difference between the current study and  Zhang et al. (2023). Please elaborate more on this.

Biao Zhang, Barry Haddow, and Alexandra Birch. 2023. Prompting large language model for machine translation: A case study

**Reasons To Accept:**

* The translation improvement is interesting and experiments are extensive to show that the improvement comes from combining multiple features together with the CTQ scorer model. The model itself is straightforward to come up IMHO, but the training proposed in the paper is really clever. I really like the method.

**Reasons To Reject:**

I vote for an acceptance of the paper. However I think there are still some comments that the paper should address:
* The experiment set up is not clear regarding how training data for the CTQ scorer is generated. It was mentioned in the paper that "Using Algorithm 1 we use the dev set of FLORES-101 (Goyal et al., 2022a), containing N = 997 sentence pairs, as the held out data along with the example database to create training data for the CTQ Scorer. We use K =100 retrieved examples per input sentence in the dev set." I am wondering why we need the dev set of FLORES-101 to generate training data as in Table 1, there are a lot of bilingual examples and we can basically split the dataset there (e.g. 997 sentence pairs as input and the rest as the datastore for collecting retrieved examples).
* Section 3.2 is very unclear what are the new features and what are not. I know there is a subsection about that in the experiments (i.e. Observations on newly proposed features), but the paper should state very clearly at Section 3.2 what are new and what are not.
* Why using BM25 to find retrieved examples results in much worse translation accuracy compared to Random selection. I found this very hard to understand and the paper does not give any discussion about this point.
* Table 6 is interesting but confusing: What if we select only the most relevant translation example as prompt instead of selecting top-4 translation examples for an input?

**Reproducibility:**

3: Could reproduce the results with some difficulty. The settings of parameters are underspecified or subjectively determined; the training/evaluation data are not widely available.

**Reviewer Confidence:**

3: Pretty sure, but there's a chance I missed something. Although I have a good feel for this area in general, I did not carefully check the paper's details, e.g., the math, experimental design, or novelty.

---

> ### Author Rebuttal · Authors · 2023-08-29
>
> Thank you for your time and encouraging review! We address your concerns below:
>
> **-Q1: Why we need the dev set of FLORES-101 to generate training data**
>
> Please note that we use the FLORES-101 dataset only as input sentences. The 100 BM25 examples per input are fetched from the larger datastore. Any parallel corpus can suffice for the input examples. We used the FLORES-101 dev set because of its quality and domain similarity to the test set, and it is sufficient to generate 99.7k examples for training the scorer. We request that our choice of FLORES-101 not be considered as a reason to reject.
>
> **-Q2: The paper should state very clearly at Section 3.2 what are new features and what are not**
>
> We will update the paper to state the new features. The new set of features we added are: chrF-InSrc, Cmt-InSrc, Cmt-InTgt, PPL-SrcTgt and, PPL-SrcTgtIn.
>
>
> **-Q3: Why using BM25 to find retrieved examples results in much worse translation accuracy compared to Random selection**
>
> Thanks for the feedback. We will add this discussion in the paper for the camera-ready version. The example database consists of some parallel sentences with just 2 or 3 tokens. And these 2 or 3 tokens might be present in the input sample and this makes the example a candidate for BM25 selection. On the other hand, a random selected example might have more tokens. In such cases, random selection outperforms BM25 selection. The learning is that such example pairs should be removed. Consequently, we're considering enhancing our filtering criteria, as referenced in lines 396 to 402, to further refine the example selection process from our database.
>
> **-Q4: What if we select only the most relevant translation example as prompt instead of selecting top-4 translation examples for an input?**
>
> We observed that for the 7B parameter model, 1-shot translation yields suboptimal performance. Since BLOOM is not heavily trained on some Indic languages, 4-shot examples are necessary to aid the model in getting good translation. That said, selecting the most relevant translation example still outperforms random selection/BM25 selection.
>
> **-Q5: The difference between the current study and Zhang et al. (2023)**
>
> Yes, ours is the first model to select examples based on a combination of features. Zhang et al. (2023) consider only individual features when selecting examples. In our work, we show that such an approach is suboptimal as it ignores the relevance of other features to the translation quality. Moreover, the selection depends on the chosen translation quality metric. So we have different metrics (e.g. COMET-QE/Labse) for the same criteria (example quality) in our selection framework. We also contribute some new features not included in Zhang et al. (2023). The new set of features we added are: chrF-InSrc, Cmt-InSrc, Cmt-InTgt, PPL-SrcTgt and, PPL-SrcTgtIn.

---

### Official Review · Reviewer_a97U · 2023-08-06

**Soundness:** 4

**Excitement:**

3: Ambivalent: It has merits (e.g., it reports state-of-the-art results, the idea is nice), but there are key weaknesses (e.g., it describes incremental work), and it can significantly benefit from another round of revision. However, I won't object to accepting it if my co-reviewers champion it.

**Paper Topic And Main Contributions:**

This paper proposes a novel approach to select few-shot examples from a parallel dataset, which are used to prompt a large language model to do machine translation. A model (CTQ) is trained to take a set of features like embedding similarity between an input source sentence and a source (or target) sentence from the dataset and produce a score indicating the potential translation quality generated by a LLM given this sentence pair from the dataset as a one-shot context. To translate a new input sentence, the dataset is searched for similar pairs using BM25 and a shortlist of candidates is reranked using the CTQ model to get top 4 pairs to be used as a few-shot context. In a set of experiments on 6 language pairs the authors show how correctly selecting few-shot examples can lead to better translation quality. They further provide a set of experiments showing the importance of different features, importance of relying on multiple features for selection, as well as experimenting with few-shot examples order and different translation quality metrics used to train the CTQ model.

**Questions For The Authors:**

- The scorer is trained on one-shot translations, but then it is used to select examples for 4-shot translation, resulting in possibly a suboptimal selection. Have you done any exploration into if this is an actual issue or not?
- It is quite surprising that example order makes such a difference on bn and ru. Presumably all top 4 matches after BM25 and scoring should be good and relevant translation pairs. Is it perhaps that the best pair is almost an exact match to the input sentence and it is best to have it last? Do you have any insight into this result?
- Just a suggestion: I understand your choice of BLOOM and XGLM for multilinguality, but I think it still would be very interesting to see results on "better" models like LLama 1/2 or ChatGPT.

**Reasons To Accept:**

- Extensive set of experiments and ablation studies
- Clear and understandable presentation
- Few shot selection is an important, but not well studied topic

**Reasons To Reject:**

- The method is somewhat clunky and expensive to run, which could limit its adoption and importance. When used with all features, this requires running a COMET model multiple times, as well as running the LLM multiple times to get perplexities, all of that before a single translation.
- [Clarification after rebuttal: this is just a suggestion, not a reason to reject] For completeness, it would have been nice to see a comparison with "cheap" finetuning methods like LoRA or soft prompting. Given the amount of processing required for computing features, it might not just have better quality, but also be more efficient.

**Reproducibility:**

4: Could mostly reproduce the results, but there may be some variation because of sample variance or minor variations in their interpretation of the protocol or method.

**Reviewer Confidence:**

3: Pretty sure, but there's a chance I missed something. Although I have a good feel for this area in general, I did not carefully check the paper's details, e.g., the math, experimental design, or novelty.

---

> ### Author Rebuttal · Authors · 2023-08-29
>
> Thank you for your time and helpful comments! We address your concerns below:
>
> **-Q1: The method is somewhat clunky and expensive to run**
>
> Yes, we acknowledge the cost of computing the features in the current scenario. However, the features can be computed in parallel and some features (like example quality) can be computed once offline - hence at least the latency can be controlled. There might be some scenarios like the need for high-quality domain specific translation where this cost is justified. We could also limit ourselves to training a scorer model on a limited set of top-k features. The key contribution of our work is to demonstrate the benefits of considering multiple factors in example selection,  and we hope that in future, better methods can reduce the cost of feature selection.
>
> **-Q2: For completeness, it would have been nice to see a comparison with "cheap" finetuning methods**
>
> Thank you for this feedback. We will do a comparison with efficient finetuning methods in the camera-ready version. It is to be noted that there might be scenarios where we do not have access to finetuning the underlying model, which is the focus of our current study. While this comparison is certainly an interesting enhancement, we request that this not be considered a reason to reject.
>
> **-Q3: The scorer is trained on one-shot translations, but then it is used to select examples for 4-shot translation**
>
> We would like to first clarify our approach again. The scorer is trained with 1 shot and then it assigns a score to each retrieved shot. This score is then used to rerank.You do raise an interesting and broader point. Ideally examples should not be selected in isolation, but should be selected to complement each other by ensuring properties like non-overlap and diversity. Selecting 4-shot examples is a subset selection problem - an NP-hard problem. We have proposed a greedy solution just as all other works have. Even single feature methods just select examples in isolation. To your immediate question, in the light of the above observations - it raises the question of how to select the 4-shots for training since it is the set of 4 candidates we are trying to score instead of one candidate. We think this is a larger subset problem that warrants further research. We have made a beginning by looking at example selection in isolation.
>
>
> **-Q4: It is quite surprising that example order makes such a difference on bn and ru**
>
> In our experiments, we observed that placing the best pair closer to the input sentence often results in a slightly better translation. Delving deeper into the examples for bn, we found that the best example, while not always an exact match to the input source, tends to have a greater number of tokens overlapping with it compared to the other three selected examples. While not an exact match, this significant overlap likely plays a role in the translation quality.
>
> **-Q5: See results on "better" models like LLama 1/2 or ChatGPT**
>
> We will consider ChatGPT for the camera-ready draft but we already tried LLama 1 translation capabilities and in our observation the translation quality is very poor for these sets of languages.

---

### Meta-Review · Area_Chair_Bpin · 2023-09-19

**Recommendation:** 4

**Metareview:**

This paper aims to improve few-shot prompting of large language language models for translation. The authors introduce a novel selection method based on a regression model using multiple feature to find better examples for the prompt. Experimental results on different language pairs leveraging different language models show the effectiveness of this approach.

The reviewers highlight the clear and understandable presentation, the extensive set of experiments as well as the comprehensive analysis and ablations. This paper could be further improved by comparing against efficient fine-tuning methods and some minor writing and presentation adjustments.

---

### Decision · Program_Chairs · 2023-10-07

**Decision:**

Accept-Findings

**Comment:**

This paper aims to improve few-shot prompting of large language language models for translation. The authors introduce a novel selection method based on a regression model using multiple feature to find better examples for the prompt. Experimental results on different language pairs leveraging different language models show the effectiveness of this approach.

The reviewers highlight the clear and understandable presentation, the extensive set of experiments as well as the comprehensive analysis and ablations. This paper could be further improved by comparing against efficient fine-tuning methods and some minor writing and presentation adjustments.